# Towards a Better Understanding of the Back-Side Illumination Mode on Photocatalytic Metal–Organic Chemical Vapour Deposition Coatings Used for Treating Wastewater Polluted by Pesticides

**Cristian Yoel Quintero-Castañeda** [1,2] , **Claire Tendero** [3] , **Thibaut Triquet** [1] , **Paola Andrea Acevedo** [2] , **Laure Latapie** [1] , **María Margarita Sierra-Carrillo** [2] **and Caroline Andriantsiferana** [1,*]

[1] Laboratoire de Génie Chimique, Université de Toulouse, CNRS, INPT, UPS, 31432 Toulouse, France; cristian.quinteroc@campusucc.edu.co (C.Y.Q.-C.); thibaut.triquet@iut-tlse3.fr (T.T.); laure.latapie@univ-tlse3.fr (L.L.)

[2] Faculty of Engineering, Universidad Cooperativa de Colombia, Santa Marta 470003, Colombia; paola.acevedop@campusucc.edu.co (P.A.A.); mariam.sierra@campusucc.edu.co (M.M.S.-C.)

[3] Centre Inter-Universitaire de Recherche et d'Ingénierie des Matériaux, Université de Toulouse, CNRS, INPT, UPS, 31432 Toulouse, France; claire.tendero@inp-toulouse.fr

[*] Correspondence: caroline.andriantsiferana@iut-tlse3.fr; Tel.: +33-5-34-32-37-03

**Abstract:** Pesticides are emerging contaminants that pose various risks to human health and aquatic ecosystems. In this work, diuron was considered as a contaminant model to investigate the influence of the back-side illumination mode (BSI) on the photocatalytic activity of $TiO_2$ coatings grown on Pyrex plates by metal–organic chemical vapour deposition (MOCVD). A photoreactor working in recirculation mode was irradiated at 365 nm with ultraviolet A (UVA) light-emitting diodes in BSI. The degradation of diuron and its transformation products was analysed by high-performance liquid chromatography, ion chromatography, and total organic carbon analysis. The coatings were characterised by X-ray diffraction analysis and scanning electron microscopy. Five coatings containing 3, 7, 10, 12 and 27 mg of $TiO_2$ exhibited different morphology, crystallinity, thickness and photocatalytic activities. The morphology and crystallinity of the coatings had no significant influence on their photocatalytic activity, unlike their mass and thickness. $TiO_2$ contents less than 10 mg limit the photocatalytic activity, whereas those greater than 15 mg are inefficient in the BSI because of their thickness. The maximum efficiency was achieved for coatings of thickness 1.8 and 2 μm with $TiO_2$ contents of 10 and 12 mg, revealing that the photocatalyst thickness controls the photocatalytic efficiency in the BSI.

**Keywords:** back-side illumination; emerging contaminants; diuron; transformation products; photocatalysis; $TiO_2$

## 1. Introduction

In the field of water treatment, emerging contaminants (ECs) are a major concern [1–3]. Even at very low concentrations (in the order of less than micrograms per litre), such contaminants and their transformation products (TPs) can have extremely harmful effects on human health and the environment [4]. Among the different types of ECs, synthetic pesticides are of particular concern [5–7] because of their potential for long-distance transport through the hydrological cycle [8–10] and their sustainability in the environment accompanied by their high toxicity and bioaccumulation capacity; hence, they pose hazards to ecosystem stability and living beings [11]. Pesticides are being increasingly used in urban areas because of their wide spectrum of applications as herbicides, fungicides and insecticides [10], and they are widely used in gardening, road and railway weed control [12]

and treatments for pets [13,14]. They are mainly detected in surface water [10,15], groundwater [16,17] and wastewater [18–20]. Wastewater treatment plants using conventional physico-chemical and biological processes may be inefficient for degrading such pollutants. For example, in various wastewater treatment plants [21–23], the herbicide diuron has been reported with very similar concentrations in the effluents (29.1–2393.1 ng L$^{-1}$) and influents (28.4–2526.1 ng L$^{-1}$). These data highlight the limitations of conventional processes for degrading ECs and the need for appropriate technologies at the tertiary treatment stage.

Among the various treatment processes adapted for degrading organic micropollutants in water, heterogeneous photocatalysis with titanium dioxide (TiO$_2$) [24], zinc oxide (ZnO) [25], nanocomposites such as tungsten oxide doped (WO$_2$/g-C$_3$N$_4$) [26], bismuth ferrite (BFO) [27] and other semiconductors have been widely studied. They generate strong oxidant radicals that react non-selectively with pollutants, initiating a series of oxidation reactions that degrade the contaminants and their TPs. Among these semiconductors, TiO$_2$ is one of the most interesting and comprehensively studied photocatalysts for degrading pesticides as it works at room temperature and pressure and has the advantages of high chemical stability, resistance to photocorrosion, high photon absorption efficiency, large specific area, low electron–hole recombination rate and low cost [24,28–30].

However, despite its excellent photocatalysis yield, TiO$_2$ has not been widely used at industrial scales because it is mostly commercially available as a nanometre-powder, which requires an additional expensive filtration step after use. This requirement makes the process expensive, tedious or even prohibitive because of the risk of carcinogenic effects [31]. To avoid the separation step required for TiO$_2$ powder, attempts were made to immobilise TiO$_2$ on a wide range of supports, such as membranes [16], activated carbon [32], metals [33] and glass [34]. These attempts involved deposition techniques such as metal–organic chemical vapour deposition (MOCVD) [35], plasma-assisted coating [36], sol–gel deposition [37] or impregnation [38]. Each of these techniques presents different challenges, such as mass transfer limitations, photocatalyst deactivation and leaching [39]. MOCVD is an excellent solution to overcome the challenges of photocatalytic coatings; nevertheless, it has a low degradation rate, which is a common problem in the case of supported photocatalysts unlike in the case of powder forms [34].

The implementation of a photocatalytic process with supported TiO$_2$ on an industrial scale is strongly limited by the photonic efficiency and lighting mode. Two modes can be considered: front-side illumination (FSI) and back-side illumination (BSI) modes [40]. FSI is the most studied mode and it has been used at the laboratory scale for research on photocatalysts; meanwhile, although BSI is of greater interest for industrial applications, it has been less studied [41]. In the FSI configuration, illumination first passes through the fluid and then radiates to the front through the photocatalytic coating [42]. This mode has the limitations that non-turbid water and shallow reactors should be used to maintain the photonic efficiency of the system; however, its advantage is that any type of support material can be used. In contrast, in the BSI configuration, the illumination first passes through the support material, then passes through the rear part of the photocatalytic coating and finally through the fluid [43]. This configuration solves the limitations of the FSI mode by allowing the use of both real turbid water and deep reactors. Its disadvantages are that only glass supports that allow passage of light can be used and that little information is available on the influence of the intrinsic characteristics of the coatings (such as crystallinity, morphology, mass and/or thickness) on the photocatalytic activity. Photocatalytic TiO$_2$ coatings under FSI have been extensively studied for different deposition techniques [33,36,44–47]. Meanwhile, only few studies have examined the BSI mode [41–43]. These studies mainly dealt with modelling the optimal coating thickness for the TiO$_2$–P25 impregnation technique.

The target molecule in this work was diuron (3-(3,4-dichlorophenyl)-1,1-dimethylurea), which belongs to the group of herbicides composed of ureas with substituted chemical groups [8]. It has been identified as a compound with carcinogenic effects [48]. Owing to its transport capacity in the aquatic environment and inhibitory action on photosystem II, it is one of the most toxic herbicides found in coastal areas, affecting the development of

corals, grasses and marine microalgae [49,50]. The degradation of diuron by heterogeneous photocatalysis with $TiO_2$ has been studied [51–54] and seems to be one of the most optimal AOPs for the complete mineralisation of the herbicide and its TPs [12,55,56]. As far as we know, no studies have reported the degradation of diuron with $TiO_2$ MOCVD coatings in the BSI mode.

Hence, with regard to photocatalytic coatings in the BSI mode, there is a lack of information on their application for water treatment and there is a need for insight into the role of the intrinsic characteristics of the coating in determining the degradation rate. In this context, the primary objective of this work was to study different $TiO_2$ coatings deposited via MOVCD and the relationship of the morphology, crystallinity and mass/thickness of the coatings with their photocatalytic activity in the BSI mode for degrading diuron present in water. After exploring these aspects, a comprehensive study was performed using the most efficient coating to explore the long-term reuse of the material, mineralisation of the pesticide and possible reaction pathways.

## 2. Materials and Methods

### 2.1. Chemicals

The diuron ($C_9H_{10}Cl_2N_2O$, ≥98%), N-demetoxylinuron ($C_8H_8Cl_2N_2O$, 95%), 1-methylurea ($C_2H_6N_2O$, 97%), 1,1-dimethylurea ($C_3H_8N_2O$, 99%), titanium tetraisopropoxide (TTIP, 97%) and powdered $TiO_2$ (Degussa P25, 99.5%) were purchased from Sigma Aldrich (Darmstadt, Germany). Solutions were prepared in distilled water. Borosilicate glass (Pyrex) substrates (138 × 18 × 3 mm) were purchased from Verre Vagner (Toulouse, France). The physico-chemical properties of diuron are summarised in Table S1.

### 2.2. $TiO_2$ Coatings by MOCVD

$TiO_2$ coatings were grown in a tubular, horizontal hot-wall reactor, as described in [57], on borosilicate glass substrates. The metal–organic precursor (TTIP) was thermally regulated in a bubbler and carried to the deposition zone with 99.999% pure nitrogen as the carrier gas. Different operating conditions were applied to obtain coatings with different characteristics (Table 1). The coating was performed in two steps, and the sample was rotated by 180° after each step to improve the homogeneity of the coating; this precaution reduced the influence of precursor depletion in the gas phase that results in a deposition rate gradient. Each coating was identified as follows: C-X, where C stands for coating and X, the rounded value of $TiO_2$ mass (in milligrams) that was deposited on the glass substrate. The samples were weighed before and after deposition to determine the mass of the coatings.

**Table 1.** Operating conditions for obtaining different coatings by metal–organic chemical vapour deposition (MOCVD) and the resulting deposited $TiO_2$ mass.

| Operating Conditions | Coatings | | | | |
|---|---|---|---|---|---|
| | **C-3** | **C-7** | **C-10** | **C-12** | **C-27** |
| Deposition time (min) | 2 × 20 | 2 × 60 | 2 × 60 | 2 × 75 | 2 × 100 |
| Precursor temperature in the bubbler (°C) | | 37 | | | 34 |
| Deposition temperature (°C) | | | 475 | | |
| Carrier gas ($N_2$) flow rate ($cm^3 min^{-1}$) | 8 | 8 | 12 | 8 | 8 |
| Dilution gas ($N_2$) flow rate ($cm^3 min^{-1}$) | | | 530 | | |
| Deposition pressure (Torr) | | | 5 | | |
| Inlet precursor mole fraction | $3.4 \times 10^{-4}$ | $3.4 \times 10^{-4}$ | $5.1 \times 10^{-4}$ | $2.8 \times 10^{-4}$ | $2.8 \times 10^{-4}$ |
| Deposited mass (mg) | 2.8 ± 0.2 | 7.5 ± 0.2 | 10.0 ± 0.2 | 12.1 ± 0.2 | 26.6 ± 0.2 |

### 2.3. Photocatalytic Degradation Experiments

All the experiments (adsorption, photolysis and photocatalysis) were conducted in the experimental device shown in Figure S1. The reactor was a 16 mL stainless-steel reactor ($12 \times 10 \times 130$ mm$^3$) with a glass window at the top. This window was located directly under a panel of monochromatic ultraviolet A (UVA) (365 nm) light-emitting diodes (LEDs) (LED Engineering, Montauban, France). Diuron solution (10 mg L$^{-1}$ and 100 mL) was circulated from an intermediate storage tank to the reactor at 200 mL min$^{-1}$ by a peristaltic pump. In this study, pH was not adjusted to fit real industrial conditions as closely as possible (without adding any chemical products). To control the temperature at 25 °C ($\pm 1$ °C), the storage tank was immersed in a thermostatic bath at 25 °C and the reactor and LEDs were placed in a continuously ventilated chamber.

The experimental protocol for the photolysis and photocatalysis experiments comprised a 60 min recirculation of the diuron solution in the dark (UV-OFF) to achieve an adsorption equilibrium and a subsequent 480 min recirculation under constant irradiation (10 mW cm$^{-2}$) of the glass window (UV-ON). Depending on the experiment, the window was made of either bare borosilicate glass (for adsorption, photolysis and photocatalysis with powder TiO$_2$–P25 (0.12 g L$^{-1}$)) or borosilicate glass with TiO$_2$ MOCVD coating (for photocatalysis with C-3, C-7, C-10, C-12 and C-27). Every 60 or 120 min, a 2 mL sample of the solution was collected from the storage tank and filtered with a 0.45 μm nylon filter. The filtered solution was analysed by high-performance liquid chromatography with UV spectroscopy and mass spectrometry (HPLC–UV/MS). The remaining solution at the end of the experiments ($\approx$85 mL) was used for TOC analysis and inductively coupled plasma optical emission spectroscopy (ICP-OES) analyses. The protocol for the C-12 coating reuse tests was the same as described above, but it included an additional step for cleaning the system by recirculating 100 mL of distilled water for 30 min between each cycle.

### 2.4. Characterisation of Materials (TiO$_2$ Coatings and Diuron)

X-ray diffraction (XRD) analysis was performed to identify the crystalline structure of the coatings. Scanning was performed using an X-ray diffractometer (GI-XRD, Bruker D8, Karlsruhe, Germany) equipped with a Cu Kα source (1.54060 Å), under a 2° grazing incidence, in the 2θ mode from 20° to 80° in steps of 0.02° and 2 s of integration time. The morphology was observed by scanning electron microscopy (SEM, LEO-435 VP-PGT, Cambridge, UK). Calowear equipment (CSEM Calotest, Dephis, Étupes, France) with a stainless-steel ball of diameter 20 mm was used to estimate the thickness of the coatings by measuring five spherical wear scars evenly distributed on the surface of each sample. The arithmetic roughness was deduced by mechanical profilometry analysis (DektakXT, Bruker, Billerica, MA, USA): 10 scans (length: 10 mm) were performed on each sample using a tip with a radius of 2 μm. Finally, the absorbance spectra of diuron were recorded by UV–Vis spectrometry (Perkin-Elmer LAMBDA 19, Norwalk, CT, USA).

### 2.5. Analytical Methods

The diuron concentrations were determined by HPLC–UV. The equipment used was a Thermo Accela HPLC-PDA (Shimadzu, Kyoto, Japan), implementing a C-18 Thermo Scientific (Waltham, MA, USA) Acclaim PA2 column (particle size: 2.2 μm, inner diameter: 2.1 mm and column length: 150 mm). Samples (10 μL) were injected into the column and UV detection was performed at 254 nm. A gradient method was used with a mobile phase composed of acetonitrile (A) ($\geq$99.9%) and ultrapure water acidified with 0.1% formic acid (B). The mobile phase gradient started at a ratio of 20%A/80%B, until it reached a ratio of 75%A/25%B in 20 min. This ratio (75/25) remained constant for 2 min and then decreased in 3 min to the initial ratio (20/80); the 20/80 ratio was maintained for another 3 min. The total analysis lasted 28 min with a constant flow rate of 0.3 mL min$^{-1}$.

Some diuron TPs were detected and quantified by HPLC-MS, while some could only be assumed to be detected. For this purpose, a Thermo Fisher UltiMate 3000 UHPLC (Sunnyvale, CA, USA) instrument connected to an Orbitrap high-resolution mass spec-

trometer (HRMS-Exactive, Waltham, MA, USA) was used. The electrospray ionisation (ESI) source was programmed in the positive mode with a capillary, tube lens and skimmer voltage of 35.00 V, a capillary temperature of 300 °C and a spray voltage of 5.00 kV. The same C-18 column and gradient method as those used in the HPLC–UV analyses were implemented. Three certified standards (N-demetoxylinuron, 1-methylurea and 1,1-dimethylurea) were used to confirm the TPs detected.

A TOC-L direct analyser (Shimadzu, Japan) was used to monitor the mineralisation of the pollutant. In each experiment, two samples were analysed—the initial and final solutions. Each sample was measured twice to determine the total carbon (TC) concentration in the first analysis and the inorganic carbon (IC) concentration in the second one. From the difference between TC and IC, the TOC concentration in the solution was determined.

$Cl^-$ ions and carboxylic acids (formic and acetic acids) were quantified by high-pressure ion chromatography (HPIC) using a Thermo Scientific ICS 5000+ ion chromatograph equipped with a conductometric detector and an AS19 anion column (4 μm, $2 \times 250$ mm). KOH (5 mM) solution was used as the mobile phase for 10 min; this was followed by an increase in the KOH concentration for 15 min to reach a value of 45 mmol $L^{-1}$. The solution was maintained at this KOH concentration for 10 min and subsequently, the KOH concentration decreased to the initial condition in 2 min. The total analysis time was 37 min, and a constant flow rate of 0.25 mL $min^{-1}$ and a temperature of 25 °C were maintained during this time.

Finally, an ICP-OES spectrometer (Ultima2, Horiba, Kyoto, Japan) was used to measure the Ti concentration in the final solution of the experiments that involved $TiO_2$ coatings. The Ti concentration was measured to investigate the strength of the coatings. The equipment has a concentric glass nebuliser and glass cyclonic spray chamber, and its operating conditions are as follows: power, 1100 W; plasma gas (Ar) flowrate, 12 L $min^{-1}$; and pump speed, 15 rpm.

## 3. Results and Discussion

### 3.1. Characterisation of $TiO_2$ Coatings

The coatings were characterised in terms of their crystalline structure, morphology, thickness and roughness by XRD, SEM, calotest and profilometry analysis, respectively, as described in Section 2.4. Figure 1 shows the XRD patterns of C-3, C-7, C-10, C-12 and C-27, indicating the crystalline structure of pure anatase $TiO_2$ for all coatings. Representative peaks were observed at 2θ values of 25.3°, 38.61°, 48.09°, 55.11°, 62.17°, 62.75°, 68.82°, 70.36°, 75.10° and 76.09° corresponding to (101), (112), (200), (211), (213), (204), (116), (220), (215) and (301) crystalline planes, respectively, in agreement with JCPDS card # 00-021-1272. The patterns did not show any peaks originating from rutile or brookite, which are the forms of $TiO_2$ that exhibit less photocatalytic activity compared to anatase [33,58].

The C-3 diffractogram presents an important background signal originating from a strong contribution of the substrate (in spite of the grazing incidence of the X-ray source) because of the low coating thickness. The relative intensities of the diffraction lines for C-10 and C-7 coatings are rather close to the ones that are referenced in the JCPDS card #00-021-1272: i.e., the (101) line being the most intense, followed by the (200), (211) and (004) lines. For the C-3 coating, a slight (211) texturation is starting to appear as (211) and (101) have similar intensities. Finally, coatings C-12 and C-27 clearly present a (211) preferential orientation and an inversion of the relative intensities of the (213) and (204) peaks.

Figure 2 shows the SEM images and roughness values of the coatings. Although all the samples are completely covered by a crack-free and uniform layer, they reveal different morphologies. C-3 presented the finest morphology with small prismatic grains because of its short deposition time and the low mass of $TiO_2$ deposited ($2.8 \pm 0.2$ mg and $2 \times 20$ min), which limit the growth of the grains. In contrast, C-10 presents the coarsest, most angular and lamellar morphology, likely because of its higher TTIP mole fraction ($5.1 \times 10^{-4}$) that causes rapid nucleation with small grains that agglomerate into larger crystallised structures owing to the constant availability of the precursor. This peculiar morphology

results in an apparent inhomogeneity of the crystalline growth that is in contrast with the growth of C-3, C-12 and C-27.

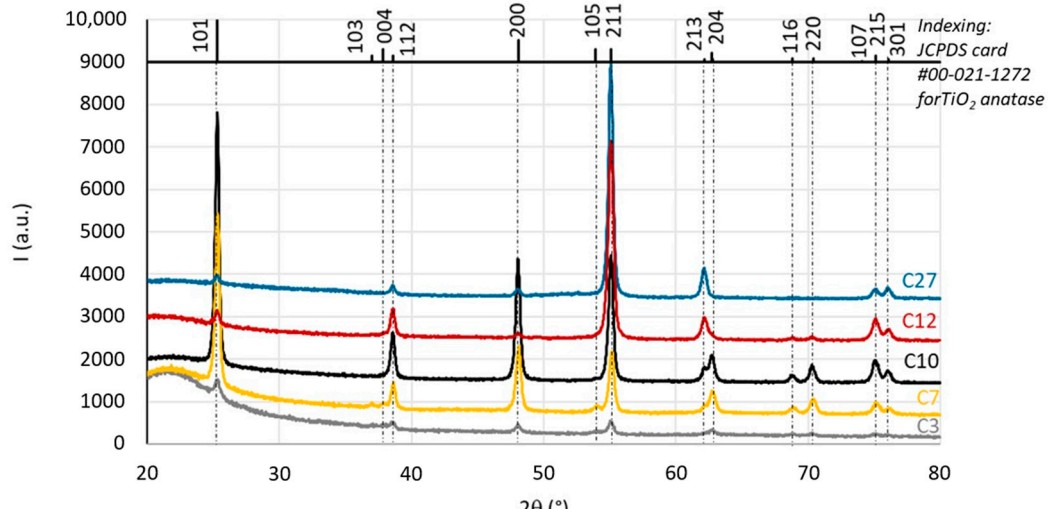

**Figure 1.** X-ray diffraction (XRD) patterns of the TiO$_2$ coatings.

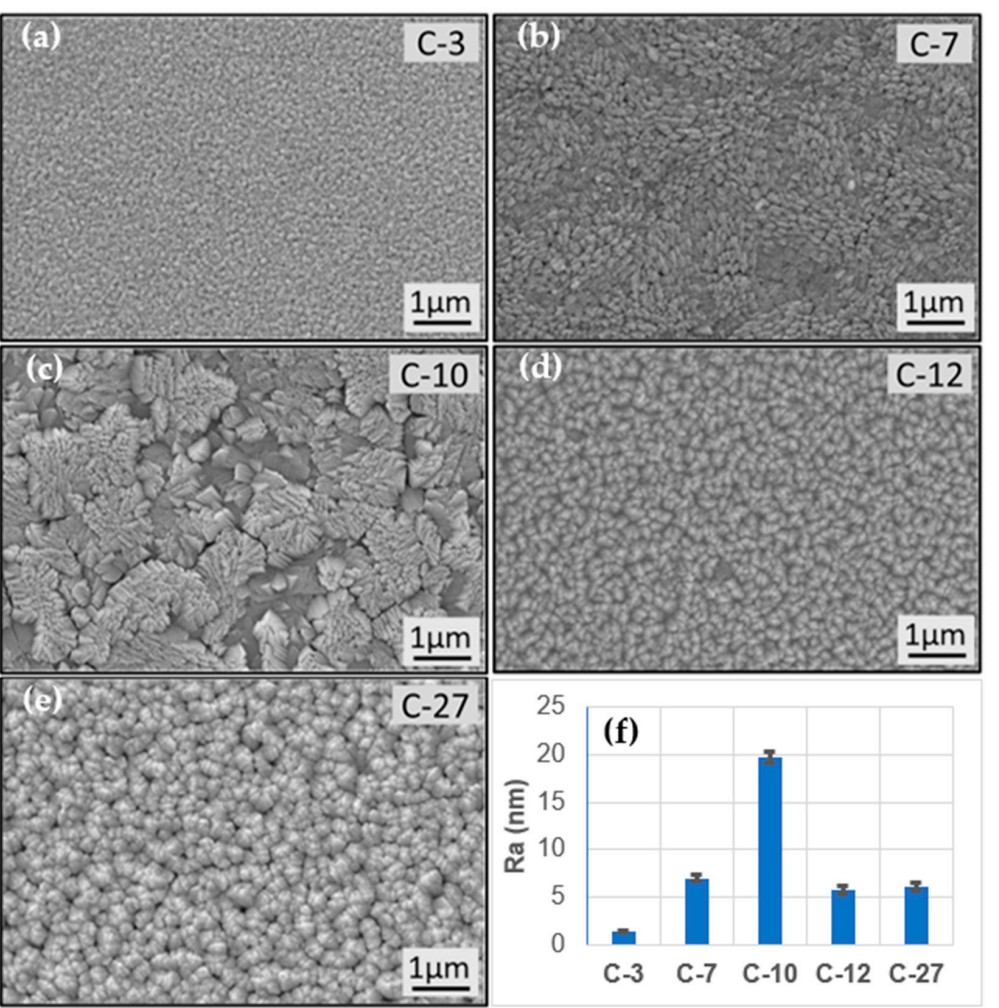

**Figure 2.** Scanning electron microscopy (SEM) image showing the top view of TiO$_2$ coatings on borosilicate glass substrates for C-3 (**a**), C-7 (**b**), C-10 (**c**), C-12 (**d**), C-27 (**e**) and arithmetic roughness (Ra) values (**f**).

C-7 also shows an apparent disorganisation with a smaller grain size. C-12 and C-27 exhibit a morphology similar to that of C-3, with evenly textured grains that become larger as the deposition time increases. Roughness values ranging from $1.4 \pm 0.1$ nm to $19.7 \pm 0.6$ nm confirm this evolution of morphology: the finest morphology corresponds to the lowest roughness and vice versa. These two types of morphology (evenly textured and apparent inhomogeneity) can be correlated with the XRD responses that show that the evenly textured grains seen in Figure 2d,e correspond to the preferential orientation of (211).

These observations are in agreement with the literature: it was previously reported that the deposition temperature (475 °C) is in the range that favours the columnar growth of the coating [59]. The grains seen in Figure 2a–e are, therefore, the heads of the columns. From the Arrhenius plot [60], we see that the growth is no longer controlled by surface reaction but by mass transfer. Therefore, the precursor mole fraction plays a major part in determining the growth. The column size and growth mode directly depend on the precursor concentration: the higher the precursor concentration, the wider are the columns [61]. This concentration also influences the column orientation.

To further highlight the influence of different morphologies on the resulting characteristics and properties of the coatings, the apparent densities were examined on the basis of thickness measurements (via the calotest, see Figure 3a) versus the mass. The results are shown in Figure 4. The plots (thickness vs. mass) of C-3, C-7 and C-12 are linearly distributed; the plot of C-27 is also linearly distributed but has a higher deviation due to higher deposition time. In a similar range of deposition time, C-10 seems slightly less dense (or more porous) than the other coatings. This deviation takes into account both measurement uncertainty and thickness gradient.

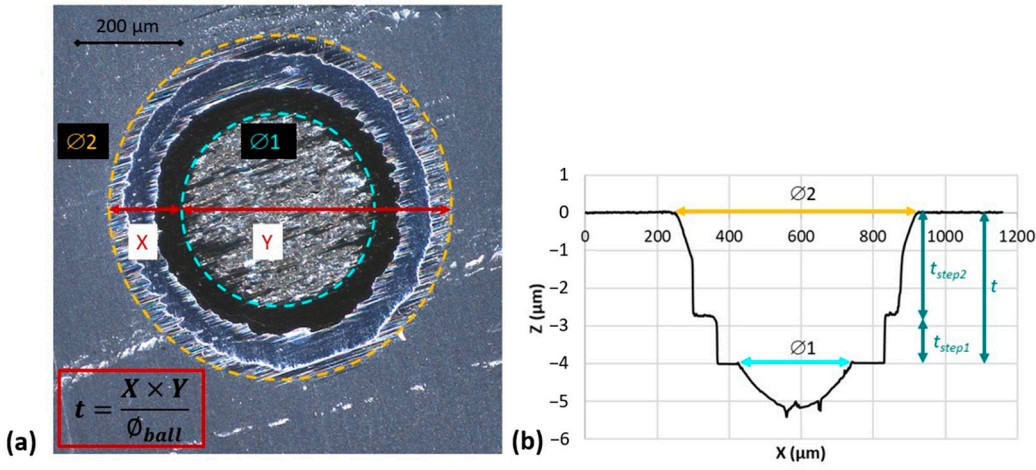

**Figure 3.** Determination of the coating thickness (**a**) by calotest and (**b**) from the crater profile measured by mechanical profilometry. In this example, $\varnothing 1 = 354$ μm, $\varnothing 2 = 670$ μm and $\varnothing$ball = 20 mm. The resulting thickness t = 4 μm, as confirmed by the profilometry measurement.

This latter gradient is highlighted by the analysis of the calotest crater by mechanical profilometry (see Figure 3b): the profile reveals the influence of the depletion of the precursor (TTIP) concentration in the gas phase that flows along the horizontal axis of the cylindrical CVD reactor, as highlighted in the literature [62]. For the example shown in Figure 3b, measurement was performed at the extremity of the substrate, where the influence of depletion of TTIP is high: in step one, the extremity is the zone farthest from the precursor introduction, whereas in step two (after the 180° rotation), it is the closest one.

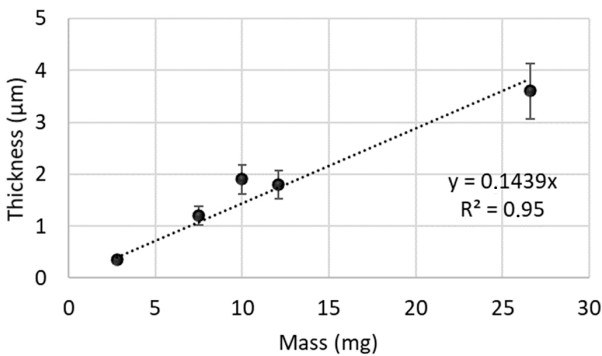

**Figure 4.** Relationship between the thickness and mass of the coatings, with stable substrate surfaces.

### 3.2. Photocatalytic Degradation and Kinetics Model

Figure 5 depicts (i) diuron adsorption inside the system, (ii) diuron photolysis (without $TiO_2$) and (iii) diuron photocatalytic degradation with $TiO_2$ (both coatings and P25 powder). Diuron elimination in the dark, without any catalyst, was 5% after 540 min. As expected, a small amount of diuron was adsorbed on the system (reactor, tubes and glass) and diuron evaporation was negligible under these experimental conditions [63]. In addition, degradation by photolysis under LEDs (UVA 365 nm) without the catalyst was low (<10% after 540 min). This result confirms that diuron is a very stable molecule and thus a problematic EC in water because it absorbs mainly in the UVC wavelength range (Figure S2). Indeed, several studies reported that the diuron half-life time (DT50) is greater than 70 days in the environment [63–65].

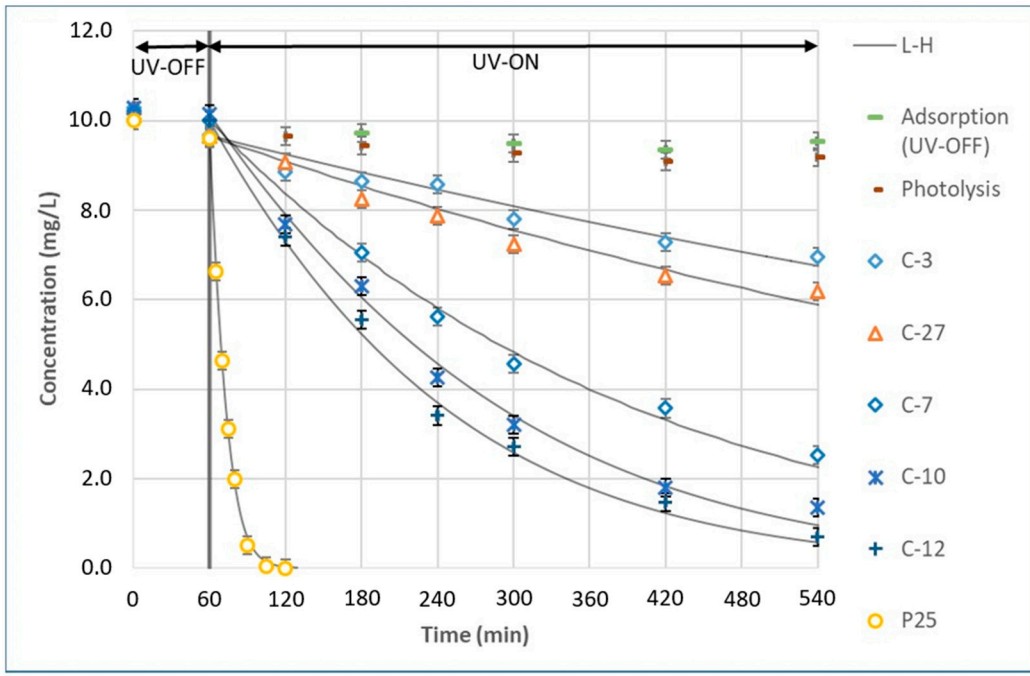

**Figure 5.** Diuron degradation kinetics for different $TiO_2$ coatings formed by metal–organic chemical vapour deposition (MOCVD) (C-3, C-7, C-10, C-12 and C-27), $TiO_2$–P25, photolysis, adsorption (without UV) and the Langmuir–Hinshelwood (L-H) model (black solid curves).

For all experiments conducted with a photocatalyst, low adsorption of diuron on both the system and catalyst (less than 6%) was observed during the first 60 min without UV light. Under irradiation, the concentration decreased at different rates showing the different performances of the catalysts. The catalysts ranked from the least to the most effective for diuron elimination (for 8 h of irradiation) are as follows: C-3 (32%), C-27 (40%), C-7 (75%),

C-10 (87%), C-12 (93%) and P25 (100%). The differences in the performances of the coatings can be explained by the differences in the amount of $TiO_2$ and their physico-chemical characteristics. Indeed, C-3 has the least $TiO_2$ (3 mg), followed by C-7 (7 mg). C-27 has the highest amount of $TiO_2$ in this study (27 mg), which is likely to screen the incident radiation. C-10 (10 mg) and C-12 (12 mg) have average amounts of $TiO_2$ that may be close to the optimal amount. Moreover, the growth of the $TiO_2$ layers by MOCVD depends on the operating conditions (temperature, precursor mole fraction, duration, etc.), which influence the morphology, texture, crystalline structure and amount of catalyst deposited. The relationship between performances and coating characteristics will be discussed in Section 3.3.

The obtained results were compared to the performances of a commercial $TiO_2$ powder P25. A concentration of 0.12 g $L^{-1}$ was selected to compare this commercial photocatalyst to $TiO_2$ coating C-12 (12.1 ± 0.2 mg $TiO_2$). C-12 required 240 min to degrade more than 50% of diuron (DT50) and 480 min to reach 90% elimination (DT90). For P25, DT50 was reached after 10 min and DT90 after 30 min; total elimination was achieved after 2 h. These results obtained with P25 agree well with the literature. Amorós-Pérez et al. [45] reported DT90 and DT50 at 30 and 10 min, respectively, for the removal of 10 mg $L^{-1}$ diuron with 0.1 g $L^{-1}$ P25. Katsumata et al. [54] reported DT90 and DT50 at 25 and 8 min, respectively, for treating 10 mg $L^{-1}$ diuron with 0.5 g $L^{-1}$ P25, while Farkas et al. [12] reported 40 and 15 min, respectively, for 40 mg $L^{-1}$ diuron and 1 g $L^{-1}$ P25. The faster degradation kinetics with the photocatalyst powder are mainly attributed to its better homogeneous dispersion compared to a thin layer of immobilised catalyst. Dispersed catalyst possibly leads to better mass transfer and diffusional gradients and promotes diuron coming in contact with the photocatalyst surface. Indeed, when using a $TiO_2$ coating and taking into account the hydrodynamics inside the reactor, a mixing problem can occur and some diuron molecules can flow through the reactor without making contact with the catalyst, possibly causing slower kinetics. Moreover, for the $TiO_2$ coatings, only one side is in contact with the liquid: that is, less photocatalyst is available for the reaction. Finally, electron–hole pairs produced by $TiO_2$ on the UV side of the glass do not manage to migrate to the liquid side. In the absence of reagents adsorbed near the catalyst ($H_2O$, $O_2$ or diuron and its TPs), the recombination phenomenon is favoured.

Equation (1) describe the Langmuir–Hinshelwood model (L-H) used to model diuron degradation by photocatalysis in the present study, where the apparent rate constant (Kapp) includes both photodegradation (k) and adsorption (Kad) phenomena [45], and C is the diuron concentration at time t.

$$-\frac{dC}{dt} = \frac{kK_{ad}C}{1 + K_{ad}C} = \frac{K_{app} * C}{1 + K_{ad}C} \text{ with } K_{app} = kK_{ad} \tag{1}$$

The L-H model was solved by means of a Matlab code using the simulanealbnd function [34]. Results are shown in Table S2. To validate the simulations, the minimised square difference between the calculated concentration (Cmod) and the experimental concentration (Cexp) was chosen as a convergence criterion ($\sum(Cmod-Cexp)^2 < 0.05$). As shown in Figure 5, the L-H model represented the concentration evolution during diuron photocatalytic degradation well. Kapp of P25 (0.1088 $min^{-1}$) was higher than that of the coatings: 0.0008 $min^{-1}$ (C-3), 0.0030 $min^{-1}$ (C-7), 0.0050 $min^{-1}$ (C-10), 0.0065 $min^{-1}$ (C-12) and 0.0011 $min^{-1}$ (C-27). These differences do not indicate the insufficiency of the photocatalytic properties of the coatings. The slower diuron degradation kinetics may be due to the limitation of the transfer under these experimental conditions. Indeed, the poor distribution of the catalyst in the reactor (as an immobilised thin layer) and the laminal flow regime (Re ≅ 300) do not favour the reaction conditions. Consequently, the kinetic constants obtained from the model are surely lower than the intrinsic kinetic constants of the coated catalyst.

The Kapp values for the C-10 and C-12 coatings indicate good performance compared to the values reported in the literature. For $TiO_2$ coatings deposited by sol–gel in the FSI

mode, Kouamé et al. [47] reported a Kapp value of 0.0025 min$^{-1}$ with an initial concentration of 10 mg L$^{-1}$ of diuron and seven times more TiO$_2$. For TiO$_2$ coatings deposited by MOCVD in the FSI mode, Kapp values of 0.06 min$^{-1}$ and 0.08 min$^{-1}$ are reported for carbamazepine and warfarin degradation [66]. For ciprofloxacin degradation in the BSI mode, a Kapp value of 0.0085 min$^{-1}$ for MOCVD TiO$_2$ coatings was reported [34].

### 3.3. Relationship of TiO$_2$ Coatings with Degradation Kinetics

First, the amount of catalyst is the key parameter that can directly influence the photocatalytic activity. This has been widely reported for photocatalytic powders [67–69]: the efficiencies increase as a function of mass up to a critical point where the overload of the system generates a shadow on the same particles and decreases the efficiency. For columnar coatings in the FSI, a similar trend has been reported [33,44,70], since the increase in mass generates taller columns and therefore increases the reaction surface. After reaching the critical point (optimal mass and/or thickness), the efficiency is either maintained constant or decreases. For our TiO$_2$ coatings, between 3 and 27 mg of TiO$_2$ were deposited with coating thicknesses of 0.3–3.6 μm. Coatings C-3 and C-27 with 3 and 27 mg of TiO$_2$, respectively, showed a trend of low photocatalytic activities (<40%). Intermediate activity (≈75%) was obtained for coating C-7. High activity (>85%) was reported for coatings C-10 (10 mg TiO$_2$) and C-12 (12 mg TiO$_2$), indicating that a critical point of maximum efficiency exists between 10 and 15 mg of TiO$_2$ (Figure 6). This critical point has been highlighted by Padoin and Soares [40], especially in photocatalytic reactors that involve BSI (i.e., the radiation first passes through the substrate, then the coating and finally the liquid).

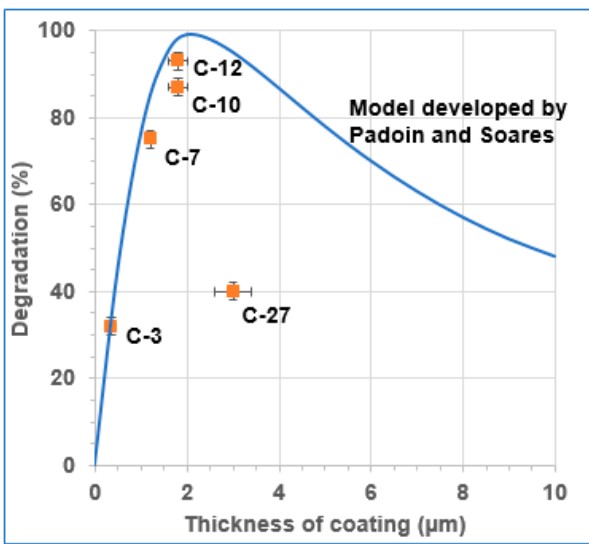

**Figure 6.** Degradation performance vs. coating thickness and the model developed by Padoin and Soares [40].

Figure 6 shows the degradation performance of each material against the coating thickness. The results show a trend that follows the model developed by Padoin and Soares [40] for TiO$_2$ photocatalytic coatings in the BSI mode (also plotted in Figure 6). As the thickness of the coating increases, the degradation efficiency increases up to the critical point (the maximum efficiency) and thereafter, it decreases. This behaviour can be explained as follows: in thin layers, the charge carrier generated in the photocatalyst may migrate to the coating/liquid interface and degrade the contaminants. As the thickness is increased up to the critical point, the absorption of the incident radiation and the production of charge carriers also increase, thereby improving efficiency. The critical point of maximum efficiency corresponds to the point with the maximum absorption of the radiation with the highest rate of transfer of the charge carriers to the coating surface. When the thickness exceeds the critical point, two phenomena may occur: radiation attenuation in the

coating and charge recombination. Radiation attenuation occurs because of the decrease in the number of photons available to generate charges near the $TiO_2$/liquid interface and charge recombination occurs because the electron–hole pairs are generated very far from the reaction surface and do not have sufficient time to migrate, thereby decreasing the photocatalytic efficiency.

The behaviours of C-3, C-7, C-10 and C-12 coatings fit the model developed by Padoin and Soares [40], while C-27 strongly deviates because of a rapid decrease in degradation efficiency. This may be explained by the fact that the model considers the photocatalytic film as a homogeneous porous matrix with variable thickness. Our coatings differ from one another in terms of column size, crystalline texture and morphology; hence, they may deviate from the model, especially in the case of thicker coatings, for which the material contribution is expected to be more important.

This deviation may suggest that the photocatalytic activity is also influenced by the crystalline structure and morphology. To further investigate this aspect and to reduce the influence of the amount of photocatalyst, we considered coatings C-10 and C-12, which have a similar thickness. This structure/property correlation is not straightforward: the two coatings have extremely different morphology and texture and yet exhibit comparable photocatalytic efficiency (>85%). Nevertheless, C-10 has a larger grain size and higher roughness, resulting in a more developed surface that promotes contact between the photocatalyst and effluent and thus improves photocatalytic activity. On the other hand, the photocatalytic efficiency is degraded in the case of (101) with crystalline texture, as seen in the case of C-10. Thus, it is possible to suggest that the influence of morphology (column size and roughness) and texture are antagonists and compensate for each other, and hence, C-10 and C-12 exhibit extremely similar photocatalytic activity.

*3.4. Mineralisation and TPs*

The TOC content was determined at the end of each experiment to evaluate the mineralisation rate (initial TOC = 5 mgC $L^{-1}$). Figure 7a shows the final TOC values for each catalyst against the initial value. To analyse these results, it is important to confirm that the total degradation of diuron and aromatic transformation products (ATPs) detected results in 58% TOC elimination (Note: For P25, the TOC was 58% at 100% degradation of diuron and its ATPs detected). If the final TOC is less than 58%, ATPs or diuron may still be present at the end of the process. The results confirmed there was no mineralisation after photolysis degradation and the TOC concentration was 5 mgC $L^{-1}$ with only 9% diuron elimination. Photocatalytic experiments with C-3 and C-27 coatings revealed 10% and 26% TOC elimination, respectively, confirming the limited performances of these coatings and the presence of reactional intermediates. Theoretically, the total degradation one mole of diuron should liberate two moles of chloride. Hence, the chloride concentration was measured simultaneously along with TOC analysis, as shown in Figure 7b. After 32% (C-3) and 40% (C-27) diuron degradation, 0.97 and 1.21 mg $L^{-1}$ of $Cl^-$, respectively, should be generated; however, at the end of the process, only 0.64 and 0.9 mg $L^{-1}$ of $Cl^-$ were detected by HPIC. This difference was caused by the incomplete mineralisation of diuron with the formation of chlorine TPs. Furthermore, $ClO^-$, $ClO_2^-$, $ClO_3^-$ or $ClO_4^-$ ions were not detected, confirming the presence of ATPs.

The use of coatings C-7, C-10 and C-12 and P25 led to higher mineralisation percentages of 38%, 44%, 52% and 58%, respectively. Photocatalytic degradation with P25 showed the total elimination of diuron; however, 42% of the TOC still remained. Moreover, except for P25, the measured $Cl^-$ concentrations for the other coatings were systematically different from the theoretical calculation. In the case of C-3 and C-27, this difference indicates the presence of aromatic and non-aromatic products.

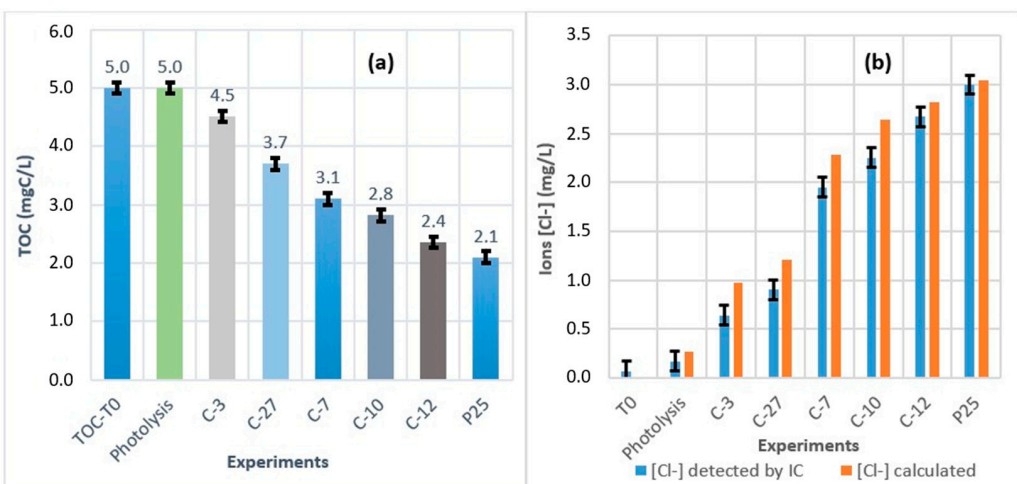

**Figure 7.** (**a**) Total organic carbon (TOC) content after 540 min of diuron degradation for different TiO$_2$ coatings formed by MOCVD. (**b**) Chloride concentrations measured at the end of the experiments and theoretically calculated from the percentage of diuron degradation.

To further understand the evolution of the TOC, the TPs generated during the photocatalytic diuron degradation were examined by HPLC–UV/MS analyses. For these analyses, we considered only C-12, which provided the best photocatalysis performance among the coatings. Table 2 summarises the information obtained by HPLC–MS to identify six TPs with a high degree of certainty.

**Table 2.** Transformation products (TPs) detected by high-performance liquid chromatography–mass spectrometry (HPLC–MS) during the degradation of diuron (DRN) with the C-12 coating.

| Molecule | Formula | Proposed Structure | Molecule | Formula | Proposed Structure |
|---|---|---|---|---|---|
| TP-1 | C$_2$H$_7$N$_2$O | | TP-2 | C$_3$H$_8$N$_2$O | |
| RT (min) LC-MS | Measured mass [M+H]$^+$ (*m/z*) | Maximum area detected | RT (min) LC-MS | Measured mass [M+H]$^+$ (*m/z*) | Maximum area detected |
| 1.62 | 75.05613 | 81,178 | 1.84 | 89.07161 | 17,049,090 |

**Table 2.** *Cont.*

| Molecule | Formula | Proposed structure | Molecule | Formula | Proposed structure |
|---|---|---|---|---|---|
| TP-3 | $C_8H_9N_2O_2Cl$ | | TP-4 | $C_9H_{11}O_2N_2Cl$ | |
| **RT (min) LC-MS** | **Measured mass [M+H]⁺ (*m/z*)** | **Maximum area detected** | **RT (min) LC-MS** | **Measured mass [M+H]⁺ (*m/z*)** | **Maximum area detected** |
| 5.36 | 201.04262 | 176,634 | 5.96 | 215.05827 | 3,193,831 |

| Molecule | Formula | Proposed structure | Molecule | Formula | Proposed structure |
|---|---|---|---|---|---|
| TP-5 | $C_9H_{10}N_2O_2Cl_2$ | | TP-6 | $C_8H_8Cl_2N_2O$ | |
| **RT (min) LC-MS** | **Measured mass [M+H]⁺ (*m/z*)** | **Maximum area detected** | **RT (min) LC-MS** | **Measured mass [M+H]⁺ (*m/z*)** | **Maximum area detected** |
| 14.60 | 249.01920 | 900,214 | 15.23 | 219.00861 | 976,775 |

The figures in Table 2 present the evolution of the concentration of each molecule with time in comparison with diuron degradation. The areas shown are normalised by the largest area detected for each molecule. The identified molecules were N-methylurea (TP-1), 1,1-dimethylurea (TP-2), 1-(3-chloro-4-hydroxyphenyl)-3-methylurea (TP-3), 3-(4-chloro-3-hydroxyphenyl)-1,1-dimethylurea (TP-4), 3-(3,4-dichlorohydroxyphenyl)-1,1-dimethylurea (TP-5) and N-demethoxylinuron (TP-6). These molecules were identified from among more than 30 diuron TPs reported in the literature [12,53,54,71,72]. The TP-2 to TP-6 molecules undergo a first phase of generation and a second phase of degradation; these phases are closely linked to the availability of the diuron molecule to generate TPs during its degradation. The maximum production of each TP occurred between 180 and 300 min of treatment. During this time, the diuron concentration started to decrease from 65% to

7%. This phenomenon can be explained by the lack of a reaction site at the surface of the catalyst until 65% diuron degradation and hydrodynamic limitation (laminar flow). At this diuron concentration, a sufficient number of reaction sites become available: the TPs in the solution are adsorbed and degraded. It is also possible that these TPs are more resistant to photocatalysis. The simultaneous degradation of TPs after more than 60% of the original compound was degraded has been reported in the case of other pesticides such as monuron and antibiotics such as ciprofloxacin [12,34]. In contrast, TP-1 followed a different trend; TP-1 was first detected 1 h later than the other TPs. TP-1 is one of the final products of the diuron reaction pathways before complete mineralisation and is generated by the degradation of the same TPs. After 180 min, the amount of TP-2 begins to decrease while that of TP-1 continues to increase, indicating TP-1 production by TP-2 demethylation (Figure S3). TP-1 was degraded in the tests with P25 and in studies performed by Calza et al. [73], confirming that this molecule can be degraded by photocatalysis with $TiO_2$. Therefore, a longer irradiation time is needed to degrade this TP. At 540 min, more than 70% of TP-2, TP-3 and TP-6 was degraded and more than 80% of TP-4 and TP-5 was degraded. This finding highlights the high performance of this photocatalytic process to degrade diuron and its TPs.

The main reaction pathways of diuron reported in the literature are the demethylation of its modified urea group and the hydroxylation and dehalogenation of the benzene ring [12,53,71,72]. For $TiO_2$ coating C-12, from the TPs identified, we can conclude that the three aforementioned mechanisms were responsible for the degradation of diuron. A schematic mechanism of the degradation is proposed in Figure S3, adapted from the literature [53,54,71].

Three TPs were quantified (PT-1, PT-2 and PT-6) with reference to certified standards (1-methylurea, 1,1-dimethylurea, N-demetoxylinuron). The results are shown in Figure S4. The final concentrations were 0.49 mg $L^{-1}$ for TP-1 and 0.07 mg $L^{-1}$ for TP-2 and TP-6, which represent the remaining TOC values corresponding to 0.16, 0.03 and 0.03 mgC $L^{-1}$, respectively, at the end of the process. Considering the remaining diuron (0.70 mg $L^{-1}$ corresponding to 0.32 mgC $L^{-1}$), the TOC contribution of these four aromatic products corresponds to 22.5% (0.54 mgC $L^{-1}$) of the final TOC (2.40 mgC $L^{-1}$). In previous works, carboxylic acids, such as formic, acetic, propionic, citric, malic and oxalic acids, were identified during diuron degradation [53,72]. In this study, only formic (1.05 mg $L^{-1}$) and oxalic acids (0.20 mg $L^{-1}$) were detected at the end of the process (the initial pH was $6.8 \pm 0.1$ and final $6.1 \pm 0.1$). Therefore, the 78% remaining TOC should correspond to the other ATPs and TPs in very low concentrations (that could not be identified or quantified) or with a low extinction coefficient at 254 nm (for UV detection in HPLC).

Most TPs detected in this study are not are hazardous substances (1-methylurea (TP1), 1,1-dimethylurea (TP2) and aliphatic acids) but some TPs like 1-(3,4-Dichlorophenyl)-3-methylurea could be as toxic as diuron [74]. Nevertheless, this study shows clearly that photocatalysis can non-selectively degrade all TPs present in the solution, provided that the irradiation time is optimised.

### 3.5. Reuse and Leaching

Four cycles of diuron degradation were performed to verify the usefulness and long-term sustainability of the coatings. For this experiment, the coating with the highest efficiency (C-12) was used. Figure 8 shows the kinetics of diuron degradation for the cycles studied with the adjustment of the L-H model and the apparent rate constant (Kapp). The coating has the same degradation efficiency for all the cycles with Kapp values of 0.006–0.007 min$^{-1}$. This is attributed to the non-selective photocatalytic degradation of all the molecules that are adsorbed on the surface of the coating, ensuring that the same reaction surface is available for all cycles. UV–Vis and XRD analysis showed that after the last cycle, the coating remained unchanged in terms of its absorbance spectrum, crystalline structure and morphology. This finding indicates that the coating was stable over time and had high reuse efficiency. In addition, in the ICP-OES analysis, the Ti in solution was

measured after each cycle to verify whether leaching occurred in the photocatalyst. Ti was not detected in solution or the concentrations were below the detection limit of the equipment (<10 µg L$^{-1}$), indicating the excellent fixation of the coating and confirming that the degradation efficiencies remained stable. The concentrations of TOC, Cl$^-$ ions and TPs of diuron remained constant in all the cycles, with values similar to those shown in Figure 7 and listed in Table 2 for C-12 coating.

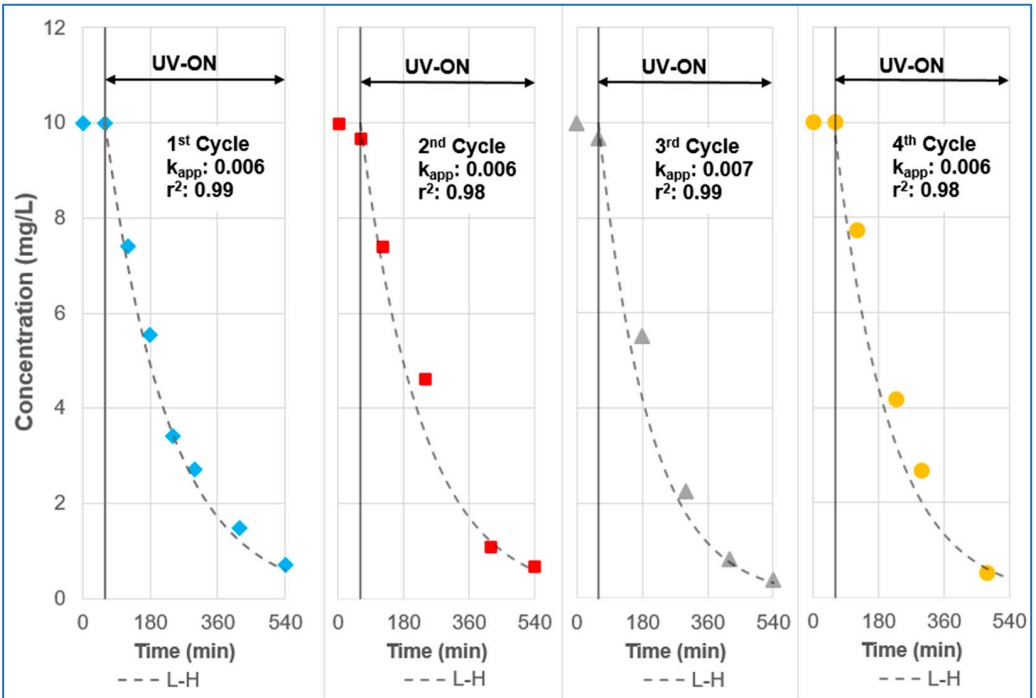

**Figure 8.** Diuron degradation kinetics for different reuse cycles of the C-12 coating and the L-H model with Kapp (min$^{-1}$).

The promising reuse efficiency of coatings formed by MOCVD has also been reported in the literature. Xingwang et al. [75] examined TiO$_2$/Fe supported on AC, and Murgolo et al. [66] coated stainless-steel meshes with TiO$_2$; both the coatings demonstrated stability and could be reused for 10 cycles of 60 min each. For sol–gel TiO$_2$ coatings, Hsuan-Fu et al. [76] reported that the photocatalyst remained stable and could be reused for four cycles of 720 min each. In contrast, He et al. [38] reported the leaching-induced deactivation of a WOx/TiO$_2$ photocatalyst supported by impregnation. According to these results, C-12 has a good reuse efficiency as leaching does not occur and the photocatalytic efficiency remains unchanged after four cycles of 540 min each (more than 2000 min). Thus, the C-12 coating is a sustainable photocatalyst.

## 4. Conclusions

Five TiO$_2$ coatings with different morphology, crystallinity and thickness/mass were grown by the MOCVD method. Their performance in the photocatalytic degradation of the herbicide diuron and its TPs in the BSI mode was examined. The amount of mass deposited determines the film thickness, which is the key parameter that directly influences the degradation kinetics in this illumination mode. Coatings C-10 and C-12 showed good photocatalytic activities for degrading diuron (>85%), and C-3 and C-27 had low activities (<40%), because of the limitations imposed by the small amount of TiO$_2$ and the electron–hole recombination and screening effect attributed to the coating thickness in the BSI mode applied. Moreover, the morphology and crystalline texture are likely to further influence the photocatalytic activity. Diuron degradation proceeded via demethylation, hydroxylation and dehalogenation. At the end of the process, mainly recalcitrant carboxylic acids (formic

and oxalic acids) that affect the remaining TOC were identified. The coatings did not experience leaching and remained stable over time, allowing their sustainable reuse with constant degradation efficiencies.

With supported catalysts, the implementation of large-scale photocatalysis is often limited by irradiation constraints: if the catalyst is deposed inside the reactor, the effluent must be transparent and the depth of the reactor low. With back-side irradiation, there is no longer a problem with photon penetration. Indeed, under these conditions, it is possible to treat turbid water and also to design reactors which deal with larger flows. This study shows that $TiO_2$ coatings grown by MOCVD present good performances for the degradation of pesticides using the BSI mode and thus open up prospects for large-scale development.

**Supplementary Materials:** The following supporting information can be downloaded at: https://www.mdpi.com/article/10.3390/w16010001/s1, Figure S1: Schematic of the experimental setup; Figure S2: Absorbance spectrum of diuron; Figure S3: Proposed reaction pathways for the photocatalytic degradation of diuron by C-12 coating: Hydroxylation (+[OH]), demethylation (-[CH_3]) and dehalogenation (-[Cl$^-$]); Figure S4: Kinetics of the formation and elimination of TP-1, TP-2 and TP-6 during diuron degradation. Table S1: Physico-chemical properties of diuron; Table S2: Kinetic constants obtained for the Langmuir–Hinshelwood (L-H) model.

**Author Contributions:** C.A., C.T., C.Y.Q.-C. and T.T. designed and supervised the experiments. C.Y.Q.-C. performed the experiments, the kinetics models, ICP-OES, TOC and HPLC–UV analysis. C.Y.Q.-C. and L.L. conducted HPLC–MS analysis. C.T. performed the $TiO_2$ compact coating on Pyrex and its characterisations by XRD, SEM, EDX and UV–Vis. C.Y.Q.-C., C.T., T.T., P.A.A., M.M.S.-C., L.L. and C.A. wrote the paper. All authors have read and agreed to the published version of the manuscript.

**Funding:** The first author's research is funded by a PhD scholarship from the Food Research Focus (Foco Alimentos) of the programme Colombia Scientific—Passport to Science (Colombia Científica—Pasaporte a la Ciencia), granted by the Colombian Institute for Educational Technical Studies Abroad (Instituto Colombiano de Crédito Educativo y Estudios Técnicos en el Exterior, ICETEX).

**Data Availability Statement:** The data presented in this study are available on request from the corresponding author.

**Acknowledgments:** The authors are grateful for the support of the projects ANR TRANSPRO and POA-INV-2790. The first was financed by Agence National de la Recherche (ANR) within the Laboratoire de Génie Chimique (Université de Toulouse, CNRS, INPT, UPS), EPOC (Université de Bordeaux, CNRS, Bordeaux INP, EPHE) and REVERSAAL (INRAE). The second was financed by the Universidad Cooperativa de Colombia—UCC within the ISI Research Group.

**Conflicts of Interest:** The authors declare no conflict of interest.

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
