# Peer review of "Towards a Better Understanding of the Back-Side Illumination Mode on Photocatalytic Metal–Organic Chemical Vapour Deposition Coatings Used for Treating Wastewater Polluted by Pesticides"

_water, doi:10.3390/w16010001_

Round 1

Reviewer 1 Report

Comments and Suggestions for Authors

The manuscript entitled “Towards a better understanding of the back-side illumination mode on photocatalytic MOCVD coatings used for treating wastewater polluted by pesticides” provides good results. Therefore, the current manuscript could be accepted for publication, but after going through a major revision.      

1.     All abbreviations should be clarified upon their first appearance in the manuscript such as (UVA LEDs) in the abstract.

2.     Could the authors provide a clarification for The relative intensities of the diffraction lines for C- 10, C7 and C-3 coatings are rather close to the reference (JCPDS card #00-021-1272), whereas coatings C-12 and C-27 clearly present a (211) preferential orientation and an inversion of the relative intensities of the (213) and (204) peaks.

3.     The authors should keep all factors constant and just change one factor at a time to clearly determine whether the formation of coarse morphology is the result of the higher TTIP mole fraction or the result of TiO2 massIn contrast, C-10 presents the coarsest, most angular and lamellar morphology, likely because of its higher TTIP mole fraction (5.1 × 10−4) that causes rapid nucleation with small grains that agglomerate into larger crystallized structures owing to the constant availability of the precursor.

4.     The SEM images and the graph of arithmetic roughness (Ra) values must be given letters (a, b, c, etc..)

5.     The graph of arithmetic roughness (Ra) values should be colored.

6.     These two types of morphology (evenly textured and apparent inhomogeneity) can be correlated with the XRD responses that show that the evenly textured grains seen in the top view SEM images correspond to (211) preferential orientation.What I understood by reading this paragraph is that C-3 has a preferential orientation to (211) because it has evenly textured grains. However, C-3 has the lowest orientation to (211) as shown in Fig 1 !!

7.      Instead of using SEM top viewsthe authors must use alphabetical order.

8.     Relationship of TiO2 coatings whit degradation kineticsit should be with.

9.     What about the toxicity of aromatic transformation products (ATPs)? Are they more or less toxic than Diuron?

10.  What is the pH level of the water sample inside the reactor?

11.  Other factors should be investigated such as different pH levels, different concentrations of Diuron.

12.  Other materials should be compared to TiO2 in the degradation of Diuron.

13.  The introduction about pesticides, their different types, their alternatives should be more enriched. Here are some suggestions that could be helpful in enriching the introduction

https://doi.org/10.1007/s13762-017-1512-y

https://doi.org/10.1016/B978-0-323-85581-5.00023-9

https://doi.org/10.1016/j.psep.2019.12.035

14.  The English language should be improved.  

Comments on the Quality of English Language

The English language should slightly improved

Reviewer 2 Report

Comments and Suggestions for Authors

Manuscript ID: water-2747253

Cristian Yoel, Quintero-Castañeda and co-authors reported “Towards a better understanding of the back-side illumination mode on photocatalytic MOCVD coatings used for treating wastewater polluted by pesticides” Although the topic is interesting, but some important aspects were not performed. Following comments should be addressed before possible consideration for publication in worthy Journal of Water.

1.      In abstract, Influence should be influence. Also check such errors throughout the manuscript.

2.      In abstract authors used abbreviations without full form. All abbreviations should be used along with full form when use first time .

3.      Mentioned % purity of all chemicals and reagents.

4.      Conclusion not clearing results and novelty of work also used rough wording.

5.      Only SEM and XRD are performed, other techniques should be performed like FTIR and TEM, EDX etc for synthesis confirmation and elemental detection.

6.      Mechanism of degradation of pesticides should be discussed graphically

7.      TiO2 coatings by MOCVD with different Methods should be discussed in introduction, also other than degradation to remove pesticides must discussed to provide better knowledge

8.      In introduction section other methods of pollutants removal should be discussed like, Surfaces and Interfaces 34 (2022) 102324, https://doi.org/10.1080/03067319.2022.2032014, Inorganic Chemistry Communications 157 (2023) 111268.

9.      Also define the importance of this removal technique for the degradation for pesticides.

10.  Graphs should be in high resolution; these all graphs are not cleared.

11.  Kinetics of degradation should be discussed

Comments on the Quality of English Language

should be improved

Reviewer 3 Report

Comments and Suggestions for Authors

The photocatalytic activity of titanium dioxide (TiO2) surface coatings of different thicknesses prepared by chemical vapor deposition of organometallic compounds was studied. Photocatalytic activity was assessed by the decomposition of 3-(3,4-dichlorophenyl)-1,1-dime- 84 thylurea (Diuron) added to water under back-side illumination (BSI), depending on the mass/thickness of the coatings and their structural features.

The photocatalytic activity of TiO2 in the form of a continuous surface coating of the internal walls of the reactor in combination with illumination from the outside through glass walls (back-side illumination) has hardly been studied, although it has a number of advantages compared to the front-side illumination mode. This work aims to fill a gap in this area.

Five coatings containing from 3 to 27 mg TiO2 with different morphology, crystallinity, thickness and photocatalytic activity were studied. Authors showed that only the mass and thickness of the coating have a significant effect on photocatalytic activity. Maximum BSI efficiency was achieved for 1.8 and 2 µm thick coatings with 10 and 12 mg TiO2 content (photocatalytic activities for degrading Diuron >85%).

The conclusions correspond to the fairly extensive research results presented on the stated topic of the work. The list of works, including 70 publications, allows us to understand the problems of the work as a whole and the achievements of the authors of the manuscript.

It should be briefly mentioned in Materials and Methods that the schematic figure of the experimental setup (photoreactor) is shown in Supporting Information.

Figure 6, 8 - insufficient contrast compared to other Figures.

Figure 5 legend - … and the Langmuir–Hinshelwood (L-H) model (black solid curves).

Round 2

Reviewer 1 Report

Comments and Suggestions for Authors

Although the manuscript has been improved, it still needs to be further improved before final acceptance.

1. The quality of Fig. 2f must be improved.

2. Why did the authors not try to investigate the effect of pH?

3. What is the type and source of wastewater used in the current study? 

Comments on the Quality of English Language

Only minor changes are required.

Reviewer 2 Report

Comments and Suggestions for Authors

Accept

Comments on the Quality of English Language

ok

Author Response

Thank you very much for taking the time to review this manuscript, for your comments and to accept our research for publication.